# Anomaly Detection on the Edge Using Smart Cameras under Low-Light Conditions

**DOI:** 10.3390/s24030772

**Published:** 2024-01-24

**Authors:** Yaser Abu Awwad, Omer Rana, Charith Perera

**Affiliations:** Department of Computer Science and Informatics, Cardiff University, Cardiff CF24 4AG, UK; ranaof@cardiff.ac.uk (O.R.); pererac@cardiff.ac.uk (C.P.)

**Keywords:** anomaly detection, low-light image enhancement, IoT-edge devices, object detection

## Abstract

The number of cameras utilised in smart city domains is increasingly prominent and notable for monitoring outdoor urban and rural areas such as farms and forests to deter thefts of farming machinery and livestock, as well as monitoring workers to guarantee their safety. However, anomaly detection tasks become much more challenging in environments with low-light conditions. Consequently, achieving efficient outcomes in recognising surrounding behaviours and events becomes difficult. Therefore, this research has developed a technique to enhance images captured in poor visibility. This enhancement aims to boost object detection accuracy and mitigate false positive detections. The proposed technique consists of several stages. In the first stage, features are extracted from input images. Subsequently, a classifier assigns a unique label to indicate the optimum model among multi-enhancement networks. In addition, it can distinguish scenes captured with sufficient light from low-light ones. Finally, a detection algorithm is applied to identify objects. Each task was implemented on a separate IoT-edge device, improving detection performance on the ExDark database with a nearly one-second response time across all stages.

## 1. Introduction

Deep learning (DL) has been involved as a key feature in many applications to replace human effort in video surveillance systems with the aid of computer vision (CV), which is one of the dominant fields concerned with extracting local and global features through the processing of digital images and live-streaming videos [1]. Traditionally, authorities relied on implementing closed-circuit television (CCTV) to monitor human–object behaviours in private and public environments. However, implementing these systems involves significant cost due to the installation procedure, the number of cameras needed depending on a particular scenario, and the need for a cloud infrastructure to record and store captured data for analysis and decision-making. Many sectors are inclined to replace traditional video surveillance systems as well as non-vision sensors with off-the-shelf cameras (also known as vision sensors) to build intelligent surveillance systems for security and other purposes.

Camera sensors are becoming increasingly accessible due to their affordability, with low cost, high resolution, and less power consumption. Moreover, they can detect, track, and identify object behaviours and send alerts in an automated manner without human interaction. However, one of the ongoing and much-discussed challenges studied in video surveillance system (VSS) domains in the last decades regards the location of processing and the analysis of the obtained data. Indeed, streaming or transmitting the acquired data to the cloud was found to entail a heavy workload on the communication network in 2017, accounting for 74% of the total network [2,3]. Many prior studies relied on processing the data on the Cloud, the Edge or both [4]. However, strategies that rely on cloud-based computing services, whether on their own or in conjunction with edge-based implementations, suffer from delay-sensitivity, bandwidth limitations and privacy and storage issues, which have crucial effects on increasing the computational cost of offloading data and the affordability of the hardware infrastructure, as well as on processing and maintaining high-volume recorded videos and still images [5].

Object detection is an essential task within the field of computer vision. It has been applied in numerous real-world situations along with video surveillance systems, including, but not limited to, autonomous driving [6] and road detection [7]. The resultant output identifies distinct instances (e.g., person, car, table, etc.) for detection and tracking objectives. However, detecting objects can be complex when operating in degraded conditions, such as inadequate night, dawn, and dusk illumination. Further, additional conditions might contaminate captured images in low-light settings with noise (e.g., salt-and-pepper effects), weather conditions (e.g., rainy, foggy, etc.), and blur. Indeed, such conditions can negatively impact object detection performance, which can become problematic for applications involving anomaly detection. Consequently, it is crucial to implement certain methodologies that possess the capability to enhance image quality and facilitate feature recovery (e.g., texture, edges, etc.). Also, it is essential for high-level tasks (e.g., object detection, segmentation, classification, etc.), where improved performance and high true positives are desired.

This research is based on the 5G Wales Unlocked project funded by the Welsh Government Department for Digital, Culture, Media and Sport (DCMS). The project utilised 5G technology to enhance different aspects of communication in rural and semi-rural regions in Wales. Different scenarios considered incorporated the integration of multiple cameras and sensors to collect data and improve understanding of the surrounding environment. The video surveillance system implemented consists of multiple cameras (Meraki MV72X) installed in four distinct locations to monitor and ensure the safety of lone workers at a farm in Monmouthshire, in the northern region of Wales. The primary goal was to identify unknown people and vehicles to avert the theft of livestock and machinery and to monitor the farmers’ activities to ensure their well-being and security. Raglan Castle in Monmouthshire was focused on to identify instances of vandalism and prevent children from scaling walls and to detect individuals trespassing in prohibited zones. The primary objective of these use cases was to detect anomalous events in either behaviour or appearance, thereby allowing for prompt human intervention and decision-making processes to ensure the safety and security of the premises. Moreover, the study focused on analyzing customer behaviour within transport services by tracking the number of individuals boarding and alighting buses and assessing the availability and occupancy of bus seats. Additionally, in Blaenau Gwent, a region located in southeast Wales, the same camera system implemented in the farm and castle scenarios was utilised to monitor parking lots, to detect available and occupied spaces, and to monitor individuals at bus-stops.

The cameras employed in all the use cases possess and provide different functionalities and capabilities, which are listed as follows: (1) A tiny machine-learning algorithm detects only people and vehicle classes along with the (a) timestamp, (b) object ID, (c) bounding box, (d) class confidence (%), (e) class name. (2) Audio level (dB), (3) Lux value, (4) A REST API to request screenshots and publish the meta-data with an (5) MQTT broker.

The data generated by the cameras (referred to as “pre-identify single”) were considered and utilised to portray any possible anomalies within the scene. When the edge camera detected such an event, a screenshot was obtained via the API and transmitted to a cloud computing environment. The cloud-based environment employed various state-of-the-art object detection algorithms to scrutinise and identify instances within the images. Afterwards, the results were shared with relevant authorities for decision-making, sending alerts, and visualisations.

As stated before, image quality plays a crucial role in producing reliable inputs for object detection tasks, leading to superior outcomes. The quality of the data source is impacted by several factors, including the communication channel, which results in the loss of some image features during data transmission, thereby degrading the image quality. However, the most significant hurdles during the project assessment pertained to the external low-light and weather conditions encountered, especially at night, as seen in Figure 1. In most cases, the lux value approached zero, producing noisy images that were challenging to interpret, especially when detecting anomalies. The subsequent section presents a comprehensive overview of diverse methodologies associated with anomaly detection. These studies primarily concentrate on tasks like detection and tracking, utilising distinct location-based computing paradigms, such as edge, edge-fog, and edge-cloud computing. Furthermore, they share a mutual objective of identifying instances involving pedestrians and vehicles in prohibited zones, the disregard of safety protocols, and violation of regulations, among other factors. However, it is worth noting that the training and evaluation datasets used in these studies were typically composed of high-quality images or videos captured in ideal conditions with ample lighting, which may lead to developing approaches that perform poorly in scenarios where the data is contaminated with noise, such as low-light conditions, also known as poor illumination.

To fill the gap in prior studies, we developed an image enhancement technique that can be employed in low-light conditions for object detection tasks on various edge devices (referred to as “nodes”). As proposed in [8], a lightweight detector and enhancer were implemented to distinguish between captured blurred and clear images. Its objective was to reinstate the sharp details if the detector considered the captured images to be blurred. In addition, the whole system was explicitly designed to adhere to the edge computing requirements. Similarly, our proposed technique was designed to select the most effective enhancement method among multiple low-light enhancement networks by considering the characteristics and features of the input images. In summary, the main contributions of this research are as follows:We developed a lightweight dynamic classifier capable of selecting the optimal enhancement technique from a set of options based on input features and illumination levels. In low-light environments, the introduction of additional noise and color artifacts poses challenges. Therefore, our classifier identifies the most effective approach during the feature extraction and selection phases, avoiding adherence to static or predetermined techniques based on assumptions.Evaluation of the efficacy of image enhancement techniques in object detection tasks involving the identification of instances such as people, cars, buses, motorcycles, etc. Specifically, we consider the most effective pre-processing stage for the proposed design by analyzing the performance of various image enhancement techniques.Evaluation of the image enhancement models on the ExDark dataset, comprising images captured in real-world scenarios. This differs from studies listed in Table 1, where models are evaluated on high-quality and paired datasets, such as SID, LOL, and MIT-5K.We developed a proof-of-concept by implementing our proposed system on a variety of resource-constrained edge-only devices, including the Raspberry Pi and Jetson Nano Developer Kit. This is in contrast to previous studies that depended on external resources, such as cloud-based solutions [9]. Our approach offers self-sufficiency on edge devices, delivering high performance by conducting measurements related to speed, detection accuracy, and other relevant computational resource metrics. The code and data to implement or test our proposed design is available at https://gitlab.com/IOTGarage/anomaly-detection-on-the-edge-using-smart-cameras-under-low-light-conditions.git (accessed on 12 November 2023).

## 2. Related Work

Anomaly detection in digital data refers to behaviours that deviate from typical ones in space, time, or both [32]. In smart city applications, anomaly detection is crucial for real-time monitoring through single or multiple cameras for inspecting vast quantities of recorded videos and still images. Researchers have exploited cloud-based services to handle computationally intensive tasks, like action recognition, tracking, etc, to identify anomalies. However, recently, several studies have investigated detecting anomalies using edge computing paradigms [32].

In [33], the authors proposed real-time video streaming for human detection and tracking by extracting low-level features using edge and fog computing for performing high-level human and action recognition tasks. Moreover, a histogram of orientation (HOG) [34] with a support vector machine (SVM) classifier [35] was used to extract object representations and to classify them as human and non-human on a Raspberry Pi 3. Finally, the features extracted were passed on to a kernelised correlation filter (KCF) [36] tracker, placed at the fog stratum (laptop) to estimate the pedestrians’ future positions and to construct their trajectories.

The authors of [37] proposed a hybrid edge-cloud computing approach that utilises a lightweight deep learning model at the edge. To reduce the complexity of the model on the NVIDIA Jetson TX, the depth-wise separable convolution technique, as introduced by [38], was employed on the convolutional neural networks (CNNs) component of both Tiny-Yolo and MobileNetV2-SSD. Meanwhile, a centralised cloud-based system with a graphics processing unit was used to host a larger model, YoloV3, which was utilised for the detection phase.

In [39], a real-time system for detecting facial emotions was devised using a PYNQ-Z1 board. Initially, a Harr-cascade algorithm and a binary neural network (BNN) were employed to extract facial features from human images and to construct a feature map. These features were then utilised for training a face emotions classifier through a BNN to detect anomalies in public transportation, including shared cabs and taxis, to ensure passenger safety. As a result, anger, disgust, fear, and sadness were categorised as abnormal, while happiness and surprise were classified as normal. On the other hand, ref. [40] described a system known as iSENSE, an intelligent surveillance system deployed at the edge. It was accomplished by developing a lightweight convolution neural network (L-CNN) based on the depth-wise separable convolution technique, as described in the work of [38], to extract human features. In addition, an SSD-head was incorporated to produce bonding boxes, probabilities, and classes. The proposed L-CNN was evaluated in conjunction with three tracking algorithms: the Karman filter (KF) [41], the kernelised correlation filter (KCF) [36], and background subtraction [42]. The system’s overall tracking performance was tested on various single-board computers (SBC), including the Raspberry Pi 3 and Tinker Board.

Furthermore, ref. [43] introduced a real-time method for performing video analytics on an NVIDIA Jeston TX2, an edge device. This approach was augmented with a graphics processing unit (GPU) to facilitate the detection of individuals in restricted areas through the utilization of YOLOv5s [44], which generates a range of object outputs, such as bounding boxes, class names, and probabilities. The DeepSORT tracker [45] was then applied to enable the tracking of detected objects and to assign a unique identification number to each object.

The above studies identified anomalies based on high-computational resources, such as cloud-based or partial (e.g., edge-cloud or edge-fog) computing frameworks. Moreover, the detection and tracking tasks were applied under favourable conditions of good visibility and ample illumination (such as daytime, sunny weather, etc.). Thus, this work involved using comprehensive operations, encompassing classification, enhancement, and detection, on several “nodes” with limited resources, utilising only the edge computing paradigm. Furthermore, this study aimed to enhance the detection phase in situations with inadequate illumination through multi-network enhancement techniques by examining the input features. Additionally, the study sought to align with the objectives outlined in the existing literature by concentrating on comparable categories, specifically humans and vehicles. The research strategy involved a deliberate focus on these particular classes to ensure relevance and consistency with prior studies in the field.

## 3. Materials and Methods

This section describes our suggested technique for enhancing images under low-light conditions for object detection. It should be noted that the primary objective of employing these methods is to accurately identify objects in poorly lit environments rather than comparing methods for image quality improvement. Consequently, the emphasis will be placed on the findings produced after the detection stage. Moreover, the performance of the state-of-the-art object detection algorithm is comprehensively assessed using a diverse set of evaluation metrics. These metrics, spanning aspects such as precision, speed, and device-related considerations, include the average precision (AP), the mean average precision (mAP), the inference speed, the number of detections, the receiver operating characteristic curve (ROC) for the classifier, the accuracy on the testing set, and metrics related to edge devices to ensure reliability and performance. This introductory overview aims to provide readers with a roadmap for navigating the detailed evaluation presented in the following sections.

The order of the following sections is determined based on the testing and evaluation phases rather than the input‘s journey from source to destination.

### 3.1. Overall System

In this subsection, we offer a description of the end-to-end proposed design, composed of three compulsory tasks; classification, enhancement and detection, as depicted in Figure 2, where the steps are described as follows:In the event of a probable anomaly (referred to as a pre-identify signal), the Meraki camera commences the process by obtaining a screenshot and transmitting it to the neighbouring node “Client (1)” for further analysis.The “Client (1)” holds a lightweight dynamic classifier in which the features are extracted and labels are assigned to denote the suitable enhancement technique based on the input’s features.A queue is designed for holding requested images and releasing them one at a time immediately after processing and completing the first image.If the images have sufficient light, $E_{0}$ is assigned, indicating that no enhancement is necessary, and they are passed immediately to the detection phase on the same node. If not, enhancement happens via $E_{1}$ or $E_{2}$ on separate nodes.The RUAS and Zero-DCE++ methods are used to enhance low-light images delivered to succeeding nodes with the labels $E_{1}$ and $E_{2}$, respectively, and are followed by the detection phase. The selected enhancement techniques are based on critical factors outlined in the subsequent section.

### 3.2. Low-Light Image Enhancement

The techniques introduced in this work vary in the type of learning on which they are based, their architecture, data training, and the framework. Table 1 illustrates the most recent accessible open-source methodologies with public implementation that have been explored between 2017 and 2021 [47]. Notably, the prevalent techniques depend on supervised learning, which involves feature mapping and metrics evaluation using paired images of the degraded and ground truths. However, as acquiring paired data of identical scenes is still a formidable task, some researchers have resorted to creating synthetic images that emulate those taken in low-light environments, which might be utilised to formulate new techniques that address the obstacles presented by low illumination levels. Nonetheless, approaches based on synthetic data exhibit poor performance in real-world situations, leading to an increase in false positives during detecting anomalies (e.g., people, vehicles, etc.), particularly at night. Consequently, researchers are exploring a new direction for addressing the low-light issue that does not rely on prior knowledge derived from reference images with normal illumination, such as zero-reference and unsupervised learning.

In the study by [13], a sub-networks enhancement MBLLEN was suggested by extracting feature representation from low-light and enhanced images for subsequent feature fusion. Conversely, [18] proposed an unsupervised learning model that employed generative adversarial learning (GAN) techniques. TheUNet [48] architecture was chosen as the generator part of the network. In general, the generator and discriminator in GAN networks fight each other until the discriminator gives up on recognising images created by the generator as fool images, producing a realistic output similar to the original ones. In contrast to most known learning approaches, [26] proposed a model without using paired images based on curve methods to create a lightweight network under the name Zero-DCE++.

Furthermore, the asterisk (*) located next to certain model names signifies a “Lightweight” architecture, which might fit constrained devices and enable faster inference. Since previous studies have not examined recently developed models, such as the CSDNet family and RED-RT, these new methods are considered and included in the evaluation with the object detection task.

Indeed, during the testing phase, several models were ignored and discarded, such as Retinex, ExCNet, and RRDNet, since these techniques perform training during inference to find the optimal value of the illumination map. Further, methods such as UTVNet and DRBN only accept paired images, which are more suitable for custom data purposes. In addition, Chen et al. and REDIIRT only accept images captured in RAW format when most modern cameras produce RBG as well as the public dataset format. On the other hand, samples should be smaller for faster processing when using KinD, KinD++, and EnlightenGAN. Indeed, after inspecting each model structure, the training data, the framework, etc., none of the models utilised the ExDark dataset (Section 3.5.2) for training and model creation; only the EnlightenGAN, Retinex-DIP, and CSDNet employed the dataset for testing, which encourages evaluation of these methods on unseen data for the detection assessment.

Ultimately, given that the primary goal is to boost the object detection accuracy, any enhancement model that outperforms the existing ones may be substituted. In other words, these models might be demonstrated to be suitable and effective as a preprocessing step for improving subsequent tasks (e.g., classification, detection, segmentation, etc.), while preserving output, quality, and latency at the edge.

### 3.3. Image Enhancement

#### 3.3.1. Zero-DCE++

Zero-reference deep curve estimation (Zero-DCE) enhances image lighting through a lightweight deep network, DCE-Net. It dynamically adjusts the dynamic range by determining specific curves for pixels and the entire image, considering factors like the pixel value range, monotonicity, and differentiability. Unique to Zero-DCE is its ability to improve images without requiring specific pairs or unrelated data during learning, thanks to innovative loss functions. This approach efficiently applies a simple nonlinear curve for effective enhancement across various lighting conditions.

#### 3.3.2. RUAS

RUAS, or Retinex-inspired Unrolling with Architecture Search, addresses the challenges of enhancing low-light images in practical scenarios without complex architectures or extensive computational resources. Inspired by the Retinex rule, RUAS develops models to understand the structure of underexposed images and unfolds optimisation processes for comprehensive enhancement. Using a cooperative reference-free learning strategy, RUAS identifies optimal architectures within a compact search space, resulting in a high-performing, computationally efficient image enhancement network. RUAS has proven superior to recent state-of-the-art methods in low-light image enhancement through rigorous experiments.

### 3.4. Object Detection

The use of a 2D object detection algorithm is an emerging field when using deep learning neural networks. The generated output is a bounding box wrapped around the detected object with numerical coordinates, confidence (%), and class name. The specifics of this output are determined by the dataset used to train these detectors and to identify desired objects. For example, most of the techniques developed in this area have been trained on image datasets, such as ImageNet [49] and MS COCO [50], which include thousands of object classes. Fine-tuning and transfer learning are popular methods to re-train the algorithm on a custom dataset or only to detect specific class name(s) [51]. One-stage and two-stage detectors are the two primary categories that make up the many types of detectors. One-stage procedures are much quicker in the training and inference stages. In contrast, two-stage procedures require more time and effort to train but provide accurate results. Thus, one common trade-off that focuses on speed against accuracy may be made when deciding between the two kinds based on the application requirements. One of the prevalent models for single-stages is known as YOLO, which stands for “You Only Look Once” and its several modifications (from Yolo to Yolov8 up to when this paper was written). At the same time, region-based convolution neural networks (RCNNs) are gaining interest for two-stage models, which include various versions, such as RCNN, Fast-RCNN, and Faster-RCNN [52].

The Faster-RCNN (Detectron2) base model with the default configuration was used for benchmarking and evaluating the low-light enhancement on the cloud paradigm [53]. Additionally, Detectron2 was exclusively employed for all use cases in the 5G Wales Unlocked project.

However, when dealing with limited resources, such as the Raspberry Pi, the primary considerations revolve around the inference speed and the model accuracy. YOLOv5 has demonstrated a trade-off between speed and accuracy in various detection applications compared to other approaches [54,55]. Moreover, it is well-suited for resource-constrained environments due to its low parameters within the model weights. Thus, in this study YOLOv5-tiny, the latest and most widely applied detection approach developed by Intel, was employed with a Neural Computing Stick 2 (NCS2) to achieve faster inference while maintaining an adequate detection accuracy on a Raspberry Pi [44,56].

### 3.5. Lightweight Image Classification

#### 3.5.1. Feature Engineering

The classification task aims to produce a label describing the content of an image, assigned with a percentage probability. Feature engineering and data-driven methods are two categories of classification techniques.

Feature engineering involves applying filters or kernels directly to the input image, resulting in a new image of the same size denoted by *f*=I⨀K, where “*I*” refers to the input image, “*K*” to the kernel, and “*f*” to the features that define and differentiate inputs based on the texture, edges, luminance, and other factors. However, selecting appropriate filters can be challenging since certain features may be difficult to identify, necessitating an extensive search of descriptors to determine the best filters and their hyper-parameters.

In contrast, modern approaches, such as convolutional neural networks (CNNs), represent a shift towards automation in the selection of kernels and their values [57]. CNNs, being part of the deep learning era, perform tasks without human intervention, making decisions on suitable kernels intuitively.

In summary, while feature engineering allows for the manual selection of filters for data-driven feature extraction, contemporary convolutional neural networks (CNNs) operate seamlessly without human intervention in the selection of kernels and their values, resulting in an increase in the model’s weight and overall complexity. Therefore, this shift towards automation positions traditional methods as being valuable for building models suitable for resource-constrained devices due to their lightweight nature, allowing for quick inference.

1.Original Pixels

In feature extraction, pixel values play a crucial role in representing the features. These values convey the original characteristics, encompassing the colour intensity in RGB or grayscale colour space, the texture, brightness, and other essential features. It is imperative to incorporate these characteristics before applying convolution kernels to ensure accurate feature extraction.

2.Gabor Filter

Gabor is a conventional tool utilised in image processing for texture analysis, edge detection, and feature extraction, specifically for image classification and segmentation purposes. It acts as a band-pass filter, enabling the transmission of specific frequencies while blocking all others, unlike high-pass and low-pass filters that solely allow the transmission of high and low frequencies, respectively. Generally, Gabor is a product of a sinusoidal signal of a certain frequency and orientation modulated by a Gaussian wave [58]. The equation representing the filter is displayed as Equation (Equation 1):(1)$$g\left(x,y,\sigma ,\alpha ,\theta ,\lambda ,\gamma ,\phi \right)=exp\left(-\frac{{\overset{´}{x}}^{2}+{\overset{´}{y}}^{2}\gamma^{2}}{2\sigma^{2}}\right)\times exp[i(2\pi \frac{\overset{´}{x}}{\lambda }+\phi )],$$
where $\overset{´}{x}$ and $\overset{´}{y}$ are expressed as:
(2)$$\overset{´}{x}=x\cos\theta +y\sin\theta$$
(3)$$\overset{´}{y}=-x\sin\theta +y\cos\theta$$

The Gabor filter relies on several parameters, mainly focused on orientation and wavelength, to govern the kernel direction and frequency. These parameters include (*x*, *y*) for the kernel size, σ for the standard deviation, θ for the kernel angle, λ for the wavelength, γ for the aspect ratio, and ϕ for the kernel offset from the original center (0, 0). The filter encompasses several parameters that produce a variety of kernels with different sizes, orientations, positions, and other factors, resulting in a multitude of filters with distinct values, collectively known as the Gabor filter bank. These filters can extract local or global features from images for subsequent tasks [59]. In addition, these parameters were subjected to value comparisons during the feature extraction phase to generate a filter bank. These filter banks are subsequently used to train a model for classification tasks. While having numerous features may seem advantageous, it is not always the case. More specific features with fewer options generally perform better.

3.Sobel Filter

The Sobel operator is a spatial domain filter that uses a convolution kernel to compute the approximate gradient of an image in the “*Gx*” and “*Gy*” directions for each pixel [60]. Unlike filters in the frequency domain, such as low-pass, high-pass, and band-pass filters (e.g., Gabor filters), which allow only specific signals to pass, the Sobel operator operates directly on the pixel values of the original image. The Sobel convolution kernel consists of two $R^{3\times 3}$ operations, as shown below:(4)$$Gx=\left[\begin{array}{ccc} -1 & \;0 & \;1\\ -2 & \;0 & \;2\\ -1 & \;0 & \;1 \end{array}\right]\;\&\;Gy=\left[\begin{array}{ccc} 1 & 2 & 1\\ 0 & 0 & 0\\ -1 & -2 & -1 \end{array}\right]$$
where the magnitude and angle are calculated as:(5)$$G_{Mag}=\sqrt{{Gx}^{2}+{Gy}^{2}},\;\theta =tan^{-1}\left(\frac{Gy}{Gx}\right)$$

The matrices above are identical in terms of their values, but one is equivalent to the other with a 90-degree rotation applied to it. The Sobel is an extension of the “Roberts” operator. However, the sole difference is the matrix’s form, denoted by the notation $R^{2\times 2}$. Further, various edge descriptors, such as the Canny, Prewitt’s, Scharr, and Laplacian operators, and various additional operators, have the same capability to identify edges in digital images. However, in contrast to the previous edge approaches, recent studies have shown that the Sobel operator outperforms other edge detection operators in preserving edges, reducing noise, and producing sharp edges [61].

#### 3.5.2. Random Forest Classifier

In the supervised learning (SL) context, Random Forest (RF) is used to predict categorical or numerical dependent variables in classification and regression problems. The RF algorithm achieves high accuracy in its predictions by constructing multiple classifiers, with a collection of decision trees (Forest) picked randomly (Random), as suggested by its name. This approach addresses the limitations of the decision tree (DT) method, which relies on a single tree for all training data, and could lead to model overfitting. By contrast, the RF algorithm builds several decision trees using several datasets, referred to as a bootstrapped dataset with various *“N”* values, in order to overcome the overfitting problem [62,63].

### 3.6. Dataset

The Exclusively Dark dataset (known as the ExDark https://github.com/cs-chan/Exclusively-Dark-Image-Dataset, accessed on 26 October 2023) comprises 7363 sample images captured with different light intensities and 13 object classes (e.g., people, cars, buses, etc.) that exhibit a range of lighting conditions from near-zero lux to partially dark [64]. These images were sourced from publicly available datasets provided by [65,66] and obtained through digital cameras and smartphones. The dataset encompasses multiple light intensity levels, providing potential feature extraction and classification utility. The ExDark dataset was extensively utilised during the enhancement and detection stage, owing to the varied lighting conditions encountered in real-world scenarios. Further, the classifier was constructed using random samples for the ExDark as well as images captured under normal light conditions from public databases, such as the Berkeley https://www2.eecs.berkeley.edu/Research/Projects/CS/vision/bsds/, accessed on 26 October 2023 [67], Stanford https://www.kaggle.com/datasets/balraj98/stanford-background-dataset, accessed on 26 October 2023 [68], MS COCO https://cocodataset.org/#download, accessed on 26 October 2023. Additionally, images from the 5G Wales Unlocked project scenarios were used to join the training and testing phases.

Addressing the low-light problem poses challenges due to the need for more suitable datasets. Many existing datasets are primarily designed to address image quality issues and lack annotation files, specifically the bounding box coordinates, which are essential for object detection tasks. Furthermore, a significant limitation arises from several datasets that predominantly feature images captured in indoor environments, rendering them less suitable for evaluating performance in real-world scenarios. Additionally, certain datasets involve images taken from behind glass or windows (e.g., inside a car), presenting a unique challenge for image enhancement techniques. Such scenarios often introduce noise and distort the content, further complicating the evaluation of these techniques. It is important to note that collecting diverse and representative data for low-light conditions proved challenging, and this consideration will be explored in future research endeavours.

## 4. Experiment

In this section, we initially assess the object detection process before and after implementing enhancements while determining essential metrics, such as the inference speed and the detection accuracy. Subsequently, we demonstrate effective techniques that yield a significant number of true positive outputs, significantly contributing to the classifier’s development. Moreover, we provide metrics in relation to edge devices when evaluating the proposed design in edge scenarios. Lastly, supplementary findings are presented in Appendix A.

### 4.1. Enhancement and Detection

#### 4.1.1. Data Preparation

The ExDark dataset includes the class name and box coordinates [l, t, w, h] generated using the Matlab toolbox [69]. Since most modern object detection algorithms generate coordinates in popular formats (e.g., COCO, Yolo, etc.), all the ground truth formats were converted to Yolo-normalised ones to facilitate the evaluation stage. Therefore, only the classes, people (with 609 images), and cars (with 638 images) were used for this part. Despite the recommended name, each dataset class contains additional instances, such as a person, car, bus, or motorcycle.

#### 4.1.2. Mean Average Precision

The primary evaluation for the object detection task is to measurethe model performance when finding objects in digital media (e.g., images and videos). Nevertheless, in some circumstances, the average precision (AP) reveals beneficial information about the detector’s performance concerning certain classes. Moreover, calculating additional metrics is essential in order to acquire the AP and mAP measures when these metrics are the precision and the recall. Following this, TP and FP are also required for estimation of the precision and recall.
(6)$$\mathrm{AP}=\frac{1}{N}*\sum_{\;}^{N}peak_{i}$$

Calculation of the average precision is described in the equation above when *N* denotes the number of interpolated points. For example, eleven points are the most commonly used when $peak_{i}\geq$ 0 represents the peak values at the *N* interpolated point in the precision versus recall plot. Afterwards, the mean average precision is obtained by the following equation, where *C* is the class name(s) [70]:(7)$$\mathrm{mAP}=\frac{AP_{person}+AP_{car}+\ldots +AP_{C}}{\mathrm{Total}\;\mathrm{number}\;\mathrm{of}\;\mathrm{classes}}$$

However, to facilitate the dataset evaluation, this research introduced an object detection metric tool suggested by [71], which allows the user to upload the following requirements: (1) the ground truth files, (2) the image files, (3) the class names file, (4) the predictions files, and (5) several options to choose the coordinates style (e.g., COCO, Yolo, and VOC), and the metrics to be measured (e.g., mAP, AP per class, $AP_{\left(50/70\right)}$ when IoU is equal to 0.5 or 0.7, and $AP_{\left(large/medium/small\right)}$ for evaluating large, medium, or small objects only). For this work, the options considered are: IOU@0.5, $AP_{\left(50\right)}$ for person, car, motorcycle and bus, mAP, and the Yolo coordinates sorted as ClassId, Confidence, X_center, Y_center, Width and Height.

#### 4.1.3. Results

All the measured metrics related to object detection before and after applying low-light image enhancement techniques are presented in this subsection. The testing and evaluation occurred in an environment with an Intel(R) Xeon Gold 6148 CPU @ 2.10 GHz (x86_64-bit) and NVIDIA Tesla P100 with CUDA 11.5. The results of the “Detectron2” baseline model on the original datasets without undergoing any enhancements are displayed in Table 2. In addition, the average detection speed on both datasets is calculated as the images are maintained at their default size and scale. Further, after the enhancement process is carried out, the metrics above are computed for each enhancement technique. In summary, the Table 3, Table 4, Table 5, Table 6, Table 7 and Table 8 presented below reveal the performance outcomes of the top five enhancement models following detection across various metrics.

Table 3 presents the average time to enhance a single image using a particular method, including the slowest and fastest times. The images were maintained at their original dimensions, which ranged from (220 and 293) to (2906 and 4372) in width and height. The superior method “Zero-DCE++” is highlighted in green, while the second-best approach and the underperforming methods for both datasets are also reported. Moreover, Table 4 shows that “Zero-DCE++” outperformed the other techniques again with a mean average precision of approximately 60% on the person dataset. In contrast, “MBLLEN” achieved an AP of approximately 50%. Moreover, “RUAS” obtained more predictions than the others. Additionally, as demonstrated in Table 5, Table 6, Table 7 and Table 8, several techniques performed well for specific classes, highlighting the importance of selecting a suitable method for conditions such as light intensities and object types. These results suggest that no single technique is universally superior and that combining different approaches is necessary for achieving optimal results for different scenarios.

Figure 3 and Figure 4 illustrate the number of predictions for the person and car datasets before and after applying enhancement techniques to the original data for the detection task. The actual regions of interest (ROIs) provided by the dataset serve as the ground truth. For example, Figure 3 shows that the detector can identify more objects as the ground truth using multiple tested models compared to the baseline model. Similarly, specific models exhibit improved performance for the car dataset. However, these additional detections may represent false predictions or accurate instances the annotators excluded during the annotation process. Nevertheless, the mean average precision (mAP) ensures that the correct ROIs align with the ground truth. The evaluation metric only compares the position and size of anticipated ROIs with those already present in the ground truth. Thus, incorrect predictions may result in overestimation or underestimation, affecting the model’s accuracy.

### 4.2. Classification

#### 4.2.1. Data Preparation and Model Selection

The study employed a total of 600 images, which were made up of 200 bright images and 200 dark images for each of the selected techniques, namely, RUAS and Zero-DCE++, since they yielded the most significant number of unique images. The augmentation involved upsampling the dark images to match the majority class (bright), seeking to balance the data by increasing the number of image features from 113 and 102 to 200, as presented in Table 9. Moreover, the data were split into train and test sets representing 90% for training and 10% for validation with a random state equal to 20. Additionally, to facilitate speedier processing and to ensure that every image was treated fairly, the dimensions of the inputted images were scaled down to 128 by 128 pixels, followed by pixel normalisation. Indeed, a series of processes, including enhancement and detection, were performed on the images, resulting in outputs with different accuracies. First, the detector’s performance on an individual input was evaluated using the average mean precision (mAP). Afterwards, the output samples with an mAP value ≥ 0.9 were included and categorised into a distinct folder representing a specific enhancement technique. At the same time, those with lower accuracies were disregarded. Because the same data samples were used for each enhancement model, unique images (outperformed by a specific enhancement technique) were isolated and evaluated individually, and duplicates were discarded; see Table 9 showing unique images with the highest mAP using a particular enhancer.

For example, the three techniques that contributed the largest number of unique samples were “Zero-DCE++”, “RUAS”, and “SLiteCSDNet_UPE”. However, due to the limited availability of ExDark samples (7 k) and a constrained number of unique outputs, only two techniques, “Zero-DCE++” and “RUAS”, were chosen to be included in the proposed edge design. In general, “Zero-DCE++” was particularly favoured and showed promising results due to its speed, overall accuracy, and accuracy in specific classes. Additionally, “RUAS” achieved the highest number of predictions compared to the ground truth and the baseline method, performing better on the “Car” class where additional predictions may reveal objects concealed by dark pixels.

#### 4.2.2. Feature Extraction and Training

As mentioned before, the “RUAS” and “Zero-DCE++” strategies were chosen to contribute to the system that was implemented at the edge for the enhancement phase. These techniques are represented by the labels $E_{1}$ and $E_{2}$, respectively. In addition, the value $E_{0}$ indicates that no enhancement is required for inputs deemed to be bright. Due to the small ExDark dataset, the number of genuinely unique samples was restricted. As a result, features were extracted using the whole datasets to represent the “Zero-DCE++” and “RUAS” by their labels, respectively.

Several filters were utilised during the extraction phase; however, only some of them contributed to the model’s ability to generate satisfactory predictions during the training phase. Examples of some of the used filters include: (1) the Gabor filter bank, (2) the Sobel, Scharr, Laplacian, and Prewitt operators for edge detection, (3) the Gaussian blur, (4) median filtering, (5) the variance filter, (6) the Sharpen filter, (7) the original pixels, and others. It is important to note that while multiple filters are suitable for data sources that use RGB and grayscale colour spaces, only filters applicable to RGB colour were introduced and applied to the collected data since the RGB colour space provides more comprehensive information for describing the entire image in various aspects. On the other hand, grayscale representation, which ranges from 0 to 255, only describes areas through black, white, or shades of grey. Therefore, for the Gabor filter configuration, it was defined with specific values: σ equal to 1 and 3, θ equal to 0 and 0.785, γ equal to 0.05 and 0.5, λ equal to 0 and 0.0785 and 1.57 and 2.356, Ksize equal to 9, and ϕ equal to 1, enabling the creation of different Gabor kernels (e.g., $G_{1},G_{2},\ldots ,G_{N}$).

On the other hand, the Sobel operator was utilised in its default configuration as well as the original pixel values for each trial conducted. After applying the convolution kernels on the input images, the features were extracted and reshaped into a 1D vector to be fed into the classifier for the training phase. The number of trees in the random forest classifier was set to *n_estimators* = 48, which was determined as the optimal value using the GridSearchCV technique to find the optimal hyperparameters with a *random_state* = 42 for all trials [72]. Furthermore, the imbalanced classes, including bright and low-light images, was solved by applying an upsampling technique to augment the features of the low-light images, given that obtaining them is relatively effortless [73].

Figure 5 illustrates that the mean test score of 93% can be attained by applying 48 trees.

#### 4.2.3. Results

Table 10 displays the accuracies of the classifiers utilised in each experiment. The features were achieved by arbitrarily utilising an assortment of filters, applying upsampling for imbalanced data, and finding the optimal number of trees using the GirdSearchCV method. The accuracy was calculated using the “metrics.accuracy_score”, which compares the predicted labels on the test set with the actual labels. As a result, the best outcomes were achieved when combining the Gabor filter ($G_{1},G_{2},G_{3}$ and $G_{4}$), the original pixel values, and the Sobel operator. The experiment (5) reported in Table 10 shows the outperformed approach surpassing the high-order kernels for the Gabor filter as well as the modern pre-trained weights convolutional neural networks (e.g., VGG-16) as a feature extractor. These features were generated using six produced kernels and then reshaped into a 1D vector. The findings demonstrated that the classifier achieved the extraction and label prediction for a single input, regardless of its random size, within 0.2 ms on the Raspberry Pi. This performance relates to real-time processing demands, ensuring a low response time at the edge with an accuracy on the test set of 85.24%, suggesting that an appropriate technique for the enhancing phase may distinguish certain low-light features as well as differentiate them from bright ones. However, it should be noted that this conclusion was reached through a single experiment on a small portion of the data. Moreover, the confusion matrix depicted in Figure 6a indicates the classifier’s ability to differentiate between bright and dark images in general and between the two chosen techniques.

The classifier algorithm encountered challenges in discriminating images exhibiting “RUAS” or “Zero-DCE++” characteristics. Figure 6b displays the receiver operating characteristic (ROC) curve which is mainly used to evaluate the performance of binary classification models. The ROC curve plots the true positive rate (TPR) against the false positive rate (FPR) at different threshold values for classification. The TPR represents the proportion of valid positive instances correctly identified as positive. In contrast, the FPR represents the proportion of valid negative instances incorrectly classified as positive [74]. The classifier obtained an ROC accuracy of 96.6%, indicating that the model can discriminate between positive and negative instances, which implies that the classifier can correctly classify positive instances while minimising false positives.

In conclusion, the images captured by the Meraki camera underwent initial processing by the classifier. During this stage, the features were extracted by utilising Gabor and Sobel filters, along with considering the original pixel values. This extraction process guided the decision-making on selecting the most suitable image enhancement based on the feature extracted.

### 4.3. Edge Computing Paradigm

In this section, the best practice enhancement models are evaluated and tested for their suitability in an edge environment. The selection of these models is based on the findings from previous benchmarking conducted on the cloud-based platform. Table 11 showcases the performance of the large model, “Detectron2” on both paradigms. The results from both datasets indicate that the application of the enhancement model leads to improved accuracy, which remains consistent even in the edge paradigm. The only notable difference is the time required to detect instances in a single image. It is important to note that batch processing of images at the edge is performed rapidly, at approximately 0.11 ms per image.

Consequently, “Yolov5-tiny” object detection was used on edge in conjunction with the enhancement models for the suggested design, as mentioned before. The test bed consisted of 40 images taken randomly from both datasets with various brightness levels (e.g., dark, semi-dark, bright, etc.) and was resized into 500 × 500 pixels. The tiny detector was evaluated using the same evaluation methods of the large model “Detectron2”. The results showed an improvement in both ‘mAP” and “AP” for both dataset classes in contrast to the baseline model when using the “RUAS” and “Zero-DCE++” techniques. Sample outputs comparing the results before and after the enhancement models were applied are as shown in Figure 6.

Furthermore, the computational resources metricised for the contributed devices, also known as “Nodes” or “Servers”, were taken into consideration throughout this work. A test of three images per category, bright, dark for $E_{1}$, and dark for $E_{2}$, was carried out in a separate experiment to evaluate device computation for a unique scenario. Table 12 shows different measurements taken simultaneously by running an input image for each class when resources such as the RAM usage, temperature readings, CPU and GPU usages were also gauged and considered for each test. The device’s measurements were obtained by creating a dashboard for a single device using a visualisation web application called “Grafana” [75]. These dashboards were then used to collect the device’s measurements and to report values by matching the timestamp during a particular process. The Grafana application works along with the“Node-Exporter” to extract the desired computation resource readings along with the “Prometheus” application to establish the connection with the dashboard and publish the metrics for visualisation [76].

For instance, the“Ideal Mode” indicates device stability, and that no processing occurs across all the nodes. On the other hand, a bright pixels image was used as the input for “Test 1”; thus, since no processing is needed, the detection job is immediately executed on the same node. In addition, it is essential to note that the CPU utilisation and temperature of “Node 3” rise due to the execution of the classification and detection tasks only. Following the same principle, “Test 2” and “Test 3” with low-light images as inputs required to have their brightness improved through“Node 1” and“Node 2” prior to the detection task. Similarly to “Node 3”, the resources increased dramatically since classification and detection were involved. For the GPU utilisation, both servers reached 99 % of usage since these enhancement models rely upon during processing.

Furthermore, an alternative design proposal was considered in the study since the evaluation results revealed high accuracy in detecting instances of “People” when employing the “Zero-DCE++” technique, achieving an average precision (AP) of approximately 0.778. Conversely, the “Car” class exhibited superior performance with an AP of approximately 0.725 when utilising “RUAS”. Therefore, both techniques can be utilised concurrently and enhance inputs in parallel since each process occurs in a separate node. Subsequently, the detection for “Car” will be conducted on images enhanced by “RUAS”, while “People” instances will be detected using “Zero-DCE++”, and the results will be combined into a single message. It is worth noting that in this situation, the same detector “Detectron2”, is recommended based on the previous evaluation. Figure 7a illustrates the detector’s failure in identifying the car without the application of any image enhancement techniques. In contrast, in Figure 7b,c, the detector successfully identifies the car using “RUAS” and “Zero-DCE++”, respectively, on the same edge devices.

The “Latency” is a vital aspect to consider since the implemented design is suggested for use on devices with limited resources; thus, it is one of the most critical variables. In other words, the whole amount of time required to perform classification, enhancement, and detection on a single picture includes the latency.

The following Table 13 and Table 14 illustrate the amount of time required to complete a separate task and the overall time across all tasks. In addition, it is interesting to note that the sole difference between RUAS and Zero-DCE++ pertains to the enhancement process, whilst all the other tasks are completed using the same approaches. Since the classification stage only resizes for label prediction, the original dimensions are passed to subsequent nodes.

## 5. Discussion

The research undertaken addressed the problem of recognising objects in environments with low-light settings at the edge. First, existing techniques for low-light image enhancement were studied along with the state-of-art object detection algorithm by evaluating the detection before and after enhancement and among multiple enhancement networks. Afterwards, best practice techniques were utilised to create a lightweight dynamic classifier that could assign labels based on image features and the amount of light intensity accumulated to choose the optimal technique for the enhancement task. Moreover, the model selection relied on the mean average precision for a single image with a value greater than 0.9, in addition to several metrics and computational resources, including the processing speed, the number of predictions, and the specific class accuracy. Finally, the end-to-end proposed system was implemented on constrained-resources devices.

According to the findings, the“Zero-DCE++” method is the optimum approach for the vast majority of situations, with regard to more rapid inference on several different input dimensions, around 0.001 s. The highest mAP was accomplished using the following techniques: “Zero-DCE++” and “MBLLEN” on the person and car datasets, respectively. The baseline model produced “mAP” values of 34 % and 45%. Both models achieved a higher “mAP” value, equivalent to 59% and 48%, than the baseline model. Again, when assessing a single class “AP”, the“Zero-DCE++” outperformed other approaches for the person, bus, and motorcycle classes with accuracies of 77%, 71% and 27%, respectively. However, the baseline results for several classes had extremely few advanced placements or almost none. The“EnlightenGAN”, on the other hand, did quite well in the car class, obtaining an“AP” score of 64%. On the other hand, the“RUAS” was the only model that produced a more significant prediction after enhancement than the others did for both datasets—about 2393 and 1836 bounding boxes, respectively—as well as compared to the actual predictions, 2073 and 1700. In addition, it was the optimal technique when it comes to detecting the“Car” classes with an AP 72%. Indeed, the results demonstrate that at least seven enhancement techniques can identify more correct objects than direct detection (without enhancement), as well as the ground truth.

Further, despite the limited samples in the ExDark dataset used to build the classifier, an accuracy of 85.24% was obtained on the test set, indicating the ability to differentiate between bright scenes from low-light scenes and among various low-light settings for an appropriate enhancement method. Indeed, the findings demonstrated close competition amongst all approaches in some facets. For example, model $E_{1}$ performed better than model $E_{2}$ on $Image_{1}$ and vice versa. Therefore, the capabilities of the various models permit the combination and integration of several models for handling various input conditions and indicate the most confident approach based on particular features within the image, making this feasible by virtue of the fact that the models can work together rather than relying on a single method, improving the object identification task.

Moreover, encouraging findings were observed in the computational resource performance of the resource-constrained devices, indicating their potential to carry out all the processing activities while ensuring high quality and rapid response times. As an illustration, the cumulative duration required to process an individual image from its origin to its destination was recorded as 0.43 s, with respective time intervals of 0.4, 0.001, and 0.03 s for classification, enhancement (utilising Zero-DCE++), and detection (employing Yolov5-tiny), indicating that other processes could be incorporated into the suggested framework, either on the same nodes or different ones. In addition, the inter-node transmission duration was under 1 s, as all the nodes are part of a shared local network.

Furthermore, since the ExDark dataset was not intended to address low-light image enhancement tasks but instead emphasised high-level tasks, such as object detection and classification, particular challenges were encountered when applying enhancement techniques to low-light images. These challenges included (1) increased coverage and obscuring of objects by dark pixels, (2) distant and small objects, and (3) partially displayed objects, where only a portion of an object is visible in the image (e.g., the rear part of a car). All these factors refer to objects not labelled and included in the ground truth. As a result, the newly discovered and missing objects considerably impact the object identification performance, leading to inadequate overall accuracy and accuracy for a specific class. Henceforth, it is suggested that the dataset be relabeled using suggested effacement techniques identified during the current research in future work, allowing other researchers to evaluate new proposed methodologies in the same field and to obtain accurate outcomes.

Regarding the classification task, the number of unique images that can outperform a single enhancement model is limited, especially during the feature extraction phase. Because of this, increasing the number of images by utilising additional datasets rather than relying solely on a single dataset for the training phase is recommended. Indeed, additional datasets other than the ExDark might boost the likelihood of obtaining more unique samples from which to extract relevant characteristics.

Furthermore, the collected metrics with regards to the resource-constrained devices across all stages pointed to a significant orientation toward consolidating more jobs and computations onto a single device. Several low-light enhancement networks may be carried out in a single “Node” when the appropriate enhancer is selected, depending on the acquired input features and attributes. Further, moving techniques related to other problems to separate nodes to the design allows for handling different challenges, such as blur and weather conditions. Thus, an input may be preprocessed and directed to numerous nodes before the detection occurs when inputs are represented with multi-labels describing the scene content. Due to logistical constraints and the targeted focus of our study on representative smart city environments, we could not conduct a comprehensive examination. We acknowledge this limitation, and future research may explore a broader range of real-world scenarios to further validate the system’s scalability and practicality.

### 5.1. Comparative Analysis with Existing Systems

Examining object recognition in degraded images within edge computing takes a secondary role compared to the processing and training of large-scale models on systems endowed with ample computational resources. This discrepancy is particularly pronounced when addressing images affected by insufficient lighting conditions. In [77], a system was proposed to enhance low-light images of electrical equipment in outdoor environments using the Zero-DCE as the enhancer technique. The study under review primarily focused on enhancement tasks and omitted additional aspects outlined in our paper, particularly object detection. It is worth noting that our work has demonstrated that the effectiveness of an object detection algorithm is largely unaffected by the image quality. As a result, higher image quality only occasionally corresponds to improved detection performance. Moreover, their approach involved utilising the IoT-Cloud, leveraging its resources for processing, managing, and storing. In contrast, our methodology encompasses implementing all methods on resource-constrained devices. Lastly, the employed Zero-DCE was compared solely with one technique, Retinex-Net. In contrast, our research is dedicated to the comprehensive evaluation of multiple techniques. Moreover, a hybrid system was devised to improve low-light images and to facilitate detection in both edge and cloud environments by [9] In this configuration, a user’s mobile device is utilised at the edge to extract the feature map. The extracted data is then transmitted through a base station to the cloud for subsequent stages of enhancement, detection, and processing. Finally, the results are sent back to the edge for further action. However, in our work, all stages, including classification, enhancement, and detection, are executed entirely on the edge, resulting in a substantially lower latency of approximately one second. In contrast to their study, the hybrid system achieved a latency of 12 s per image due to the involvement of cloud processing. The localisation of all the processing stages at the edge in our methodology contributes to the notable reduction in latency, enhancing the system’s overall efficiency. Additionally, our methodology outperforms theirs in detection accuracy, achieving approximately 70%, compared to their 62.2%, showing the effectiveness of our edge-based model in balancing low latency and high accuracy across tasks.

### 5.2. User Privacy and Data Security

Ensuring adherence to privacy and data security guidelines is imperative in all surveillance systems, especially systems that rely on partial processing between edge and cloud computing paradigms. This includes defining privileges for accessing the data, obscuring faces in images, and implementing hashing and cryptography techniques on the transmitted data [78]. However, each system may have distinct criteria tailored to the specific project requirements and scenarios. Our system considers only the specific images containing detections at the edge devices, transmitting them through an API to the cloud for subsequent processing, while undesired data are discarded, ensuring that the devices adhere to privacy compliance standards. Additionally, this approach leads to limited data processing, as only specific images undergo the processing and transmitting stages. In addition, images containing detections, such as vehicle and people, are exclusively accessible to authorised authorities and stakeholders.

Furthermore, image enhancement techniques are designed to enhance various aspects of image quality, yet their real-world implementation frequently does not match their performance in controlled settings. In this study, we demonstrated the improved efficiency of detecting several classes after enhancement compared to the pre-enhancement stage. However, the results also underscored deficiencies in image quality, particularly affecting the recognition of faces and vehicle license plates. Interestingly, this limitation may inadvertently address privacy concerns by making it harder to identify individuals or sensitive information within the images. Furthermore, the positioning of cameras during installation plays a critical role in ensuring data quality, impacting the distinct recognition of features. For instance, in the 5G Wales Unlocked project, cameras are installed at elevated positions, posing a particular challenge in distinguishing features and concurrently contributing to privacy preservation.

## 6. Conclusions

In this research, a technique was developed to improve the detection accuracy of objects by enhancing images captured under low-light conditions. Initially, the input images were subjected to feature extraction. Subsequently, a distinctive label was assigned to identify the most effective model from multiple enhancement networks. Each task was carried out using a separate resource-constrained device. The findings illustrated an improvement in accuracy when enhancement techniques occur before the detection phase by boosting the object detection accuracy by more than 20%. Moreover, each technique was superior under particular conditions, for object classes, images captured with noise accumulated, weather conditions, and resolutions, indicating that diverse capabilities might be obtained from a single technique to handle a specific situation. Furthermore, the proof-of-concept indicated the practicability of deploying the entire system exclusively with the edge computing paradigm without depending on external computational resources achieving 1 s from source to destination, consistent with real-time requirements and encouraging the addition of more tasks on the same node as well as additional nodes to address further challenges.

## Figures and Tables

**Figure 1 sensors-24-00772-f001:**
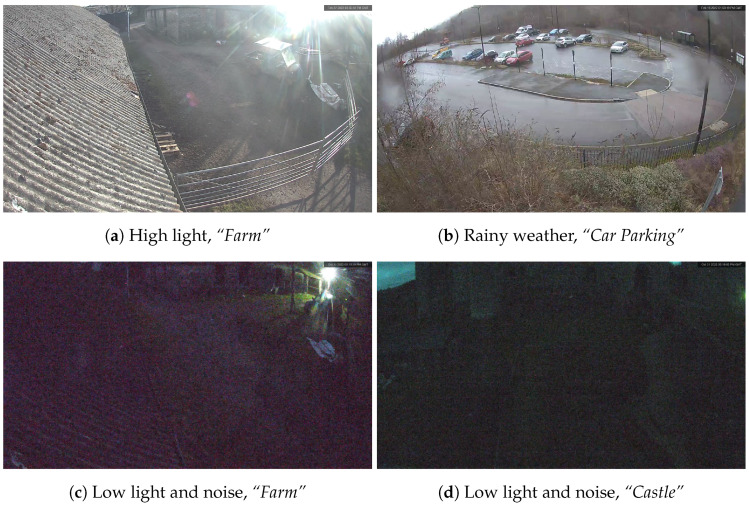
The 5G Wales Unlocked project scenarios with various natural conditions with several representative instances. (**a**) Vehicle in the farm scenario with bright lighting from a sun source partially covering it. (**b**) Depiction of the car park scenario during rainy weather, producing finely-detailed imagery. (**c**,**d**) Images captured in the farm and castle scenarios exhibiting poor illumination and significant noise.

**Figure 2 sensors-24-00772-f002:**
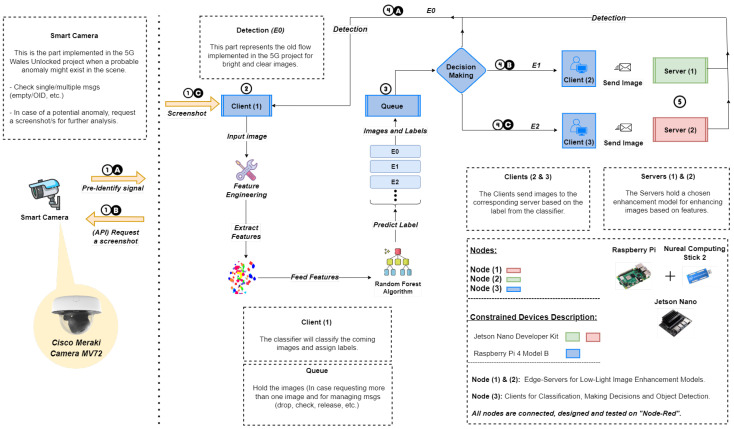
Our proposed end-to-end system design. We illustrate the proposed design from source to destination for detecting anomalies on multiple constrained devices when images captured by the edge camera are enhanced through multiple low-light image techniques, donated by {$E_{1}$, $E_{2}$,…, $E_{N}$}, based on the inputs’ local and global features. All nodes were designed, connected, and tested using the “Node-Red” platform [46].

**Figure 3 sensors-24-00772-f003:**
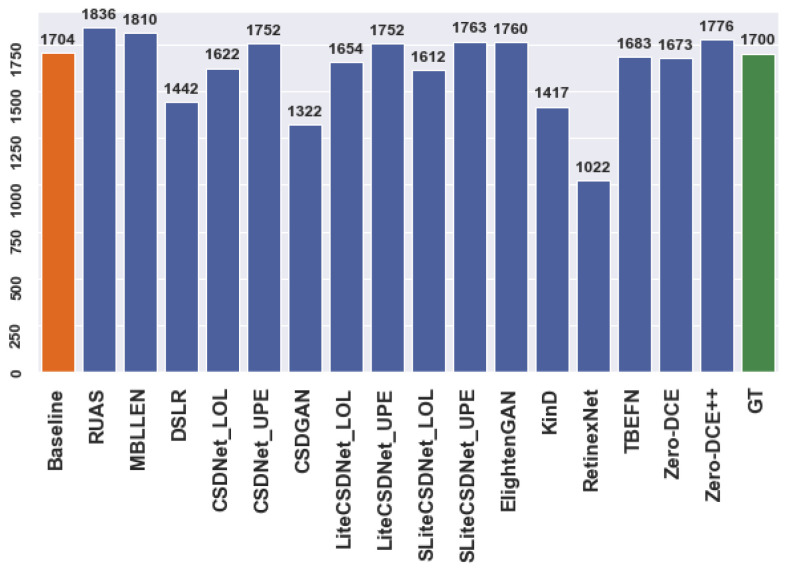
The number of ROIs, “Car” dataset.

**Figure 4 sensors-24-00772-f004:**
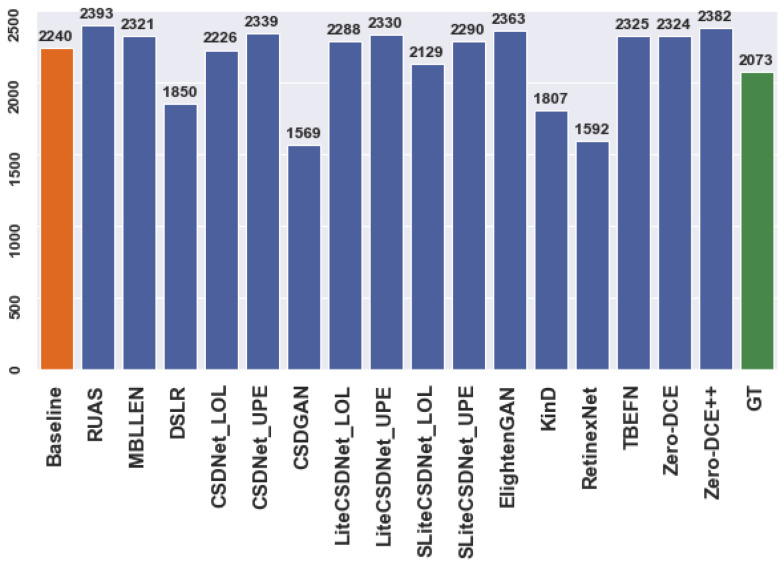
The number of ROIs, “Person” dataset.

**Figure 5 sensors-24-00772-f005:**
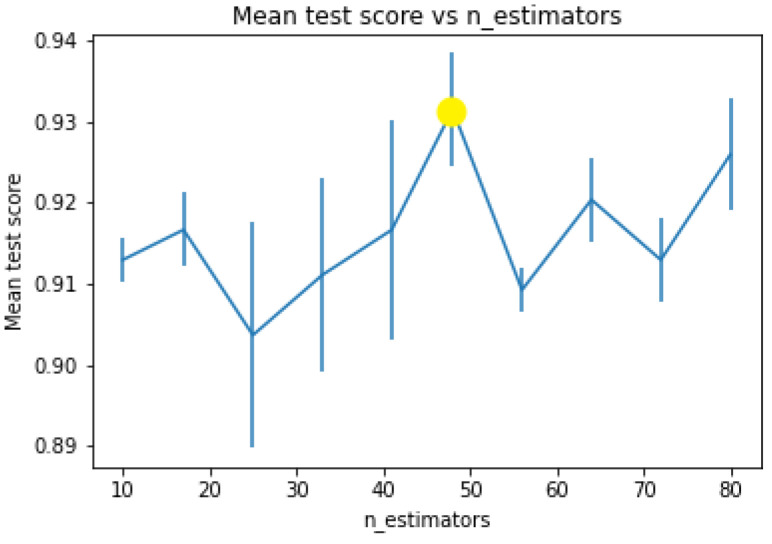
The plot represents the n_estimators in a range of 10 to 80 with the mean test scores. It can be noted that the best score highlighted in yellow is achieved when trees = 48, obtaining a score of 93%.

**Figure 6 sensors-24-00772-f006:**
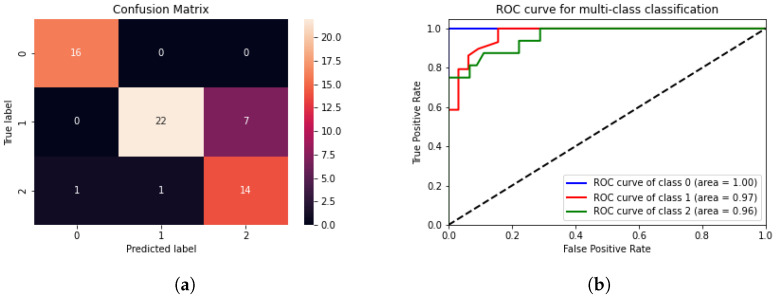
(**a**) Confusion matrix, “0” for Bright, “1” for the RUAS model, and “2” for the Zero-DCE++ model. (**b**) ROC curve for the multi-classification.

**Figure 7 sensors-24-00772-f007:**
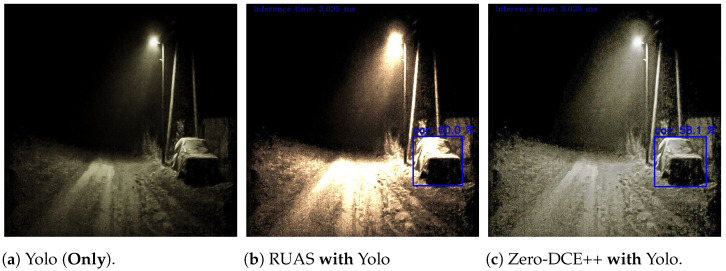
Detection outputs on the constrained devices using the state-of-art Yolov5-tiny object detection algorithm and the chosen enhancement techniques.

**Table 1 sensors-24-00772-t001:** Learning types for low-light image enhancement techniques, SL: supervised learning, USL: unsupervised learning, SSL: semi-supervised learning, ZSL: zero-shot learning.

Method/Learning	SL	USL	SSL	ZSL
LLNet [10]	✓			
LightenNet [11]	✓			
RetinexNet [12]	✓			
MBLLEN [13]	✓			
Chen et al [14]	✓			
DeepUPE [15]	✓			
KinD [16]	✓			
KinD++ [17]	✓			
EnlightenGAN * [18]		✓		
ExCNet [19]				✓
Zero-DCE [20]				✓
DRBN [21]			✓	
Xu et al. [22]	✓			
TBEFN [23]	✓			
RRDNet [24]				✓
DSLR [25]	✓			
Zero-DCE++ * [26]				✓
RUAS * [27]	✓			
Retinex-DIP [28]				✓
UTVNet [29]	✓			
CSDNet [30]	✓			
CSDGAN [30]		✓		
LiteCSDNet-LOL * [30]	✓			
LiteCSDNet-UPE * [30]	✓			
SLiteCSDNet-LOL * [30]	✓			
SLiteCSDNet-UPE * [30]	✓			
RED-RT * [31]		✓		

**Table 2 sensors-24-00772-t002:** Baseline metrics (before enhancement).

Dataset Class	Num. of Predictions	Person (AP %)	Car (AP%)	Bus (AP%)	Motorcycle (AP%)	mAP (%)	Time (ms)	Num. of GT
Car	1704	0.115	0.438	0	0	0.1386	0.12	1700
Person	2240	0.1885	0.1415	0.25	0	0.145	0.12	2073

**Table 3 sensors-24-00772-t003:** The fastest and slowest enhancement model. Bold indicates the fastest model.

Model Name	Inference Speed (s)
ElightenGAN	6.38
**Zero-DCE++**	**0.0013**
SLiteCSDNet_LOL	0.0024
TBEFN	5.77
Zero_DCE	0.0023

**Table 4 sensors-24-00772-t004:** The high and low “mAP”. Bold indicates model with highest mAP.

Model Name	Average Precision
RetinexNet	0.300
MBLLEN	0.587
**Zero-DCE++**	**0.590**
SLiteCSDNet_UPE	0.486
CSDGAN	0.370

**Table 5 sensors-24-00772-t005:** The high/low AP for “Person” class. Bold indicates model with highest ${\mathrm{AP}}_{Person}$.

Model Name	Average Precision
CSDGAN	0.524
RetinexNet	0.569
CSDNet_UPE	0.696
**Zero-DCE++**	**0.778**
RUAS	0.760

**Table 6 sensors-24-00772-t006:** The high/low AP for “Car” class. Bold indicates model with highest ${\mathrm{AP}}_{Car}$.

Model Name	Average Precision
LiteCSDNet_LOL	0.504
**RUAS**	**0.725**
ElightenGAN	0.639
MBLLEN	0.633
CSDGAN	0.475

**Table 7 sensors-24-00772-t007:** The high/low AP for “Bus” class. Bold indicates model with highest ${\mathrm{AP}}_{Bus}$.

Model Name	Average Precision
All Models	0
**Zero-DCE++**	**0.504**

**Table 8 sensors-24-00772-t008:** The high/low AP for “Motorcycle” class. Bold indicates model with highest ${\mathrm{AP}}_{Motorcycle}$.

Model Name	Average Precision
SLiteCSDNet_UPE	0.194
LiteCSDNet_LOL	0.225
**Zero-DCE++**	**0.277**
ElightenGAN	0.052
MBLLEN	0.247

**Table 9 sensors-24-00772-t009:** Unique samples with mAP ≥ 0.9 (in bold) after applying image enhancement techniques and detection on the whole 7k ExDark dataset.

Model Name	Number of Unique Images	Time (s)
CSDNet_UPE	33	0.005
LiteCSDNet_UPE	23	0.0035
LiteCSDNet_LOL	52	0.0032
**RUAS**	**113**	**0.12**
SLiteCSDNet_UPE	59	0.002
**Zero_DCE++**	**102**	**0.0012**

**Table 10 sensors-24-00772-t010:** Details of the conducted experiments with different convolution kernels and VGG-16 CNN (as a feature extractor), where OP, GB, US, and HP stand for the original pixels, the Gabor bank filter, upsampling, and the hyperparameters, respectively. Bold indicates the classifier with highest accuracy.

Exp ID	Filters/Model	US	HP	Accuracy
1	OP, GBF and Sobel	✗	✓	85%
2	OP, GBF and Sobel	✓	✗	81%
3	VGG-16	✓	✗	75%
4	VGG-16	✓	✓	78%
5	**OP, GBF and Sobel**	**✓**	**✓**	**85.24%**

**Table 11 sensors-24-00772-t011:** Evaluation of the large model “Detectron2” on the Cloud and Edge computing paradigms, w: with enhancement and wo: without enhancement.

Env	mAP%	P (Person)	AP (Car) %	Speed (in s)	Size
Cloud (wo)	0.42103767	0.625056609	0.638056402	0.09	500 × 500
Cloud (w)	0.425116397	0.573534606	0.701814585		
Edge (wo)	0.42103767	0.625056609	0.638056402	4.4	500 × 500
Edge (w)	0.425116397	0.573534606	0.701814585		

**Table 12 sensors-24-00772-t012:** Nodes computational resource metrics, where N/A: Not applicable, N/U: Not used in processing, C: Classification task, OD: Object detection task, IE: Image enhancement task, Node 1: Enhancement model (1), Node 2: Enhancement model (2), and RPi: Raspberry Pi.

	CPU Usage			GPU Usage		RAM Usage			Temp Usage					Task Involved		
Test ID	Rpi (Node 3)	Jetsons		Jetsons		Rpi (Node 3)	Jetsons		Rpi (Node 3)	Jetsons				Rpi (Node 3)	Jetsons	
		Node 1	Node2	Node 1	Node2		Node 1	Node2		Node 1		Node2			Node 1	Node 2
										CPU	GPU	CPU	GPU			
Idle Mode	1.47%	7.23%	14.50%	0%	0%	800 MB	1.23 GB	1.28 GB	41 °C	29 °C	27.5 °C	27 °C	25.5 °C	N/U	N/U	
**1**	15.00%	N/A		N/A		1.15 GB	N/U		49.9 °C	N/U		N/U		C + OD	N/U	
**2**	14.70%	N/U	65.60%	N/U	13%	1.19 GB	N/U	3.16 GB	50 °C	N/U		31 °C	29.5 °C	C + OD	N/U	IE
**3**	17.20%	31%	N/U	98%	N/U	1.13 GB	2.68 GB	N/U	49.2 °C	36 °C	33.5 °C	N/U		C + OD	IE	N/U

**Table 13 sensors-24-00772-t013:** Total time taken for processing one-image (from source to destination) using “RUAS”.

Node Id	Task	Time (in s)
1	Classifier	0.4
2	Enhancement	0.03
3	Detection	0.03
Total Time:		0.46

**Table 14 sensors-24-00772-t014:** Total time taken for processing one-image (from source to destination) using “Zero-DCE++”.

Node Id	Task	Time (in s)
1	Classifier	0.4
2	Enhancement	0.001
3	Detection	0.03
Total Time:		0.431

## Data Availability

No new data were generated in this work.

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
