# Peer review of "Anomaly Detection on the Edge Using Smart Cameras under Low-Light Conditions"

_sensors, 2024, doi:10.3390/s24030772_

Round 1
Reviewer 1 Report
Comments and Suggestions for Authors
1.There are some grammar errors. The 'on processing and maintain' on page 2 should be corrected to 'on processing and maintaining.' The 'see Figure 1' on page 3 should be corrected to 'as seen in Figure 1.'
2.In section 4.4.1 on page eight, you primarily discussed the differences between traditional methods and modern methods in selecting kernels and their values, where the latter does not require human intervention, while the former necessitates an extensive search of descriptors. However, the final conclusion drawn is that traditional methods are helpful in building models suitable for resource-constrained devices. The connection between the two is somewhat abrupt; you may need to provide additional context or rephrase it.
3.This study utilized a variety of evaluation metrics, such as inference speed, mAP (mean Average Precision), AP (Average Precision) for specific categories, the number of ROIs (Region of Interest), unique sample count, ROC accuracy, and even included RAM usage and temperature index. These metrics are scattered throughout the paper without an overall summary at the beginning of the Materials and Methods section, making it challenging for readability and comprehension.
4.The "Materials and Methods" section should focus on describing the methods you adopted. In page eight, you introduced various object detection algorithms, but the description of the Yolov5-tiny, which you ultimately chose, is only one sentence long. You should provide additional details on why you selected Yolov5-tiny and its advantages compared to other algorithms.
5.On page 14, you mentioned that not all of the filters used during the extraction phase had a positive impact. Specifically, which ones were not effective? Why were they still used instead of being omitted? You should provide a clearer explanation or conduct ablation experiments to determine the impact of each filter on the results.
6.The AP for the "person" category described on page 17 for "RUAS" and the AP for the "car" category for "Zero-DCE++" are inconsistent with Tables 5 and 6. Please recheck the data for accuracy.
7.You ultimately selected "RUAS" and "Zero-DCE++" as low-light enhancement models, but these two models did not perform the best on all evaluation metrics. Additionally, as you mentioned, "Zero-DCE++" achieved the highest values in mAP and AP for the person, bus, and motorcycle categories, while "RUAS" achieved the highest values in the number of ROIs and AP for the car category. Why were the evaluation metrics not consistent in the selection of these two models? You should restate the reasons for choosing both of them.
Comments on the Quality of English Language
Minor editing of English language required
Author Response
You will find all the responses in the attached file.

Reviewer 2 Report
Comments and Suggestions for Authors
In this study, the author developed a technique to improve the detection accuracy of objects by enhancing images captured under low light conditions. In response to the issues in the manuscript, the following suggestions are made before receiving it.
1. In terms of method description in the manuscript, the author provided a detailed description of the components used, but did not provide a good description of their contributions and innovations. Therefore, it is suggested that the core of this section should be to describe one's own innovative points and methods.
2. Would it be better to describe the dataset in the experimental section? Of course, this may also depend on the author's writing habits.
3. The professional name or method abbreviation that first appears in the manuscript should be appropriately labeled with the source, such as "RUAS" and "Zero DCE++" in the manuscript
4. The images displayed in the manuscript are almost the results of existing methods. It is recommended that the author add some images that reflect the performance advantages of the system described in the manuscript.
5. Some sentences have grammatical errors, and some sentences are not coherent.
Comments on the Quality of English Language
Minor editing of English language required.
Author Response

(The authors gave the same response as above.)

Reviewer 3 Report
Comments and Suggestions for Authors
The paper "Anomaly Detection on the Edge using Smart Cameras under Low-Light Conditions" presents a novel technique for enhancing images in low-light conditions to improve object detection in smart city surveillance systems. It integrates feature extraction, classification, and advanced object detection algorithms, addressing the challenges of monitoring outdoor urban and rural areas under poor visibility. The study employs the Exclusively Dark dataset for evaluation. It demonstrates improved detection performance with nearly one-second response time, offering significant potential for real-world surveillance and safety monitoringapplications. The paper significantly contributes to the smart city surveillance field. However, there are areas where improvements could enhance its scientific rigor and practical applicability:
Areas for Improvement:
Dataset Diversity and Size: The study heavily relies on the Exclusively Dark dataset. Expanding the dataset to include more diverse lighting conditions and environments could provide a more robust evaluation of the system's effectiveness.
Algorithmic Transparency and Interpretability: While the paper outlines the process of object detection and image enhancement, it could benefit from a more in-depth explanation of the algorithms and their decision-making processes, especially for the classifiers and enhancement techniques.
Comparative Analysis with Existing Solutions: The paper should include a more comprehensive comparison with current state-of-the-art systems. This would contextualize the proposed system's performance relative to other models and highlight its unique contributions.
Real-World Testing and Scalability: The paper would benefit from testing in a broader range of real-world scenarios to assess the scalability and practicality of the system in different smart city environments.
Handling of Varied Anomalies: The paper focuses on object detection under low-light conditions, but it could also explore how the system performs in detecting a wider range of anomalies, particularly in complex urban environments.
Performance Metrics and Evaluation: While the paper presents a thorough evaluation of the system, additional metrics such as system latency, computational efficiency, and energy consumption could provide a more comprehensive understanding of its practical viability.
User Privacy and Data Security: In the context of smart city surveillance, addressing how the system handles user privacy and data security is crucial. The paper could benefit from discussing these aspects to align with ethical guidelines and privacy regulations.
The paper should undergo Major Revision. The core concept and preliminary results are promising, demonstrating a novel approach to improving smart city surveillance under challenging low-light conditions. However, addressing the above areas would significantly enhance the study's depth, clarity, and practical relevance, ensuring a more comprehensive understanding of its implications and effectiveness in real-world scenarios.
Author Response

(The authors gave the same response as above.)

Round 2
Reviewer 1 Report
Comments and Suggestions for Authors
Detail comments:
1. material and method description can give more details. For example: dataset size, and data collection weather...
2. In 4.3. Image Enhancement, some technology descriptions can be refined. (the current description is too long)
3. The motivation section can be merged into the introduction part. Thus the readers could easily follow.
Comments on the Quality of English Language
English is OK
Author Response
Please find my responses in the attached file.
Many Thanks,
Yaser,

Reviewer 3 Report
Comments and Suggestions for Authors
The authors addressed all my concerns satisfactorily. Its accepted from my side.
Author Response
Please find my response in the document attached.
Many Thanks,
Yaser,
